# Is Obesity a Risk or Protective Factor for Open-Angle Glaucoma in Adults? A Two-Database, Asian, Matched-Cohort Study

**DOI:** 10.3390/jcm10174021

**Published:** 2021-09-06

**Authors:** Wei-Dar Chen, Li-Ju Lai, Kang-Lung Lee, Tzeng-Ji Chen, Chia-Yen Liu, Yao-Hsu Yang

**Affiliations:** 1Department of Ophthalmology, Chang Gung Memorial Hospital, Chiayi 33305, Taiwan; weidar1023@gmail.com (W.-D.C.); lynnlai@cgmh.org.tw (L.-J.L.); 2College of Medicine, Chang Gung University, Taoyuan 33302, Taiwan; 3Department of Radiology, Taipei Veterans General Hospital, Taipei 11217, Taiwan; miguelkllee@gmail.com; 4School of Medicine, National Yang Ming Chiao Tung University, Taipei 11221, Taiwan; tjchen@vghtpe.gov.tw; 5Department of Family Medicine, Taipei Veterans General Hospital, Taipei 11217, Taiwan; 6Health Information and Epidemiology Laboratory, Chang Gung Memorial Hospital, Chiayi 33305, Taiwan; qchiayen@gmail.com; 7Department of Traditional Chinese Medicine, Chang Gung Memorial Hospital, Chiayi 33305, Taiwan; 8School of Traditional Chinese Medicine, College of Medicine, Chang Gung University, Taoyuan 33302, Taiwan

**Keywords:** obesity, open-angle glaucoma, risk factor, young adults

## Abstract

Obesity contributes to multiple systemic disorders; however, extensive discussion regarding obesity and open-angle glaucoma (OAG) remains limited, and conclusions in the existing literature diverge. This study aims to analyze the risk of OAG among obese adults in Taiwan. In this study, adults (aged ≥18 years) with a diagnostic code of obesity or morbid obesity registered in the Longitudinal Health Insurance Database (LHID) 2000 and LHID2005 from 1 January 2001 to 31 December 2010 were included. All adults were traced until the diagnosis of OAG, the occurrence of death, or 31 December 2013. Risk of OAG was significantly higher in obese adults than in non-obese adults after multivariable adjustment (adjusted hazard ratio (aHR): 1.43 (95% confidence interval (CI) 1.11–1.84)/aHR: 1.54 (95% CI 1.23–1.94) in the LHID2000/LHID2005). Both databases demonstrated that young obese adults (aged ≤40 years) had a remarkably increased risk of OAG compared with young non-obese adults (aHR 3.08 (95% CI 1.82–5.21)/aHR 3.81 (95% CI 2.26–6.42) in the LHID2000/LHID2005). This two-database matched-cohort study suggests that obese adults have an increased risk of OAG. In young adults, in particular, obesity could be a potential risk factor of OAG.

## 1. Introduction

Obesity represents one of the biggest health emergency issues worldwide. According to the World Health Organization, the prevalence of obesity (body mass index (BMI) ≥ 30 kg/m^2^) in the United States was 36.2% in 2016 [1]; this figure was projected to soar to 50.7% by 2030 [2]. Similarly, the prevalence of obesity in Europe and Asia has increased exponentially over the past decade. Obesity has a multifactorial association with the environment, dietary habits, sedentary lifestyles, and genetics [3]. Obesity-derived metabolic dysregulation, inflammatory stress, and neural degeneration lead to a series of pathophysiological processes [4]. Numerous studies have indicated that obesity has a strong connection with diabetes mellitus, hypertension, ischemic heart disease, stroke, and Alzheimer’s disease [5,6].

Although discussing the association between obesity and sight-threatening disorders has recently intensified [7], the relationship between obesity and open-angle glaucoma (OAG) remains controversial. OAG, the most prevalent subtype of glaucoma globally, is characterized by progressive damage of retinal ganglion cells (RGCs), enlarged optic disc cupping, and irreversible deterioration of the visual field. Lowering intraocular pressure (IOP) is considered to be the most effective strategy in preventing disease progression, although the etiology of OAG has been not completely elucidated.

To date, several studies have reported that obesity has a positive correlation with increased IOP [8,9,10]. Nevertheless, whether obesity is a risk factor for OAG remains inconclusive [11,12]. In the Rotterdam study, there was no significant association between BMI and OAG after multivariable adjustment [13]. However, Newman-Casey showed that obese patients had a 14% increased risk of OAG in the univariable analysis, and obese women had a 6% increased hazard for OAG after multivariable adjustment [14]. Conversely, Pasquale reported that increased BMI was associated with a 6% lower risk of OAG in Caucasian women [15]. Additionally, Kim et al. showed that overweight status is a protective factor of OAG [16]. Prompted by inconsistent reports of the association between obesity and OAG, the present 13-year matched-cohort study aimed to analyze the risk of OAG among obese individuals using data from two Taiwanese population-based longitudinal databases.

## 2. Materials and Methods

### 2.1. Databases

In this study, participant data were collected from the Longitudinal Health Insurance Database (LHID) 2000 and LHID2005, which belong to the National Health Insurance research database (NHIRD) in Taiwan. The National Health Insurance program is a universal and compulsory health insurance program benefiting >99.6% of the Taiwanese population to date, and data of the NHIRD, including registration files and original claim data, have been used in many studies with high academic value. The LHID2000 contains 1,000,000 de-identified insurance claim data randomly retrieved from the year 2000 Registry of Beneficiaries of the NHIRD (23.75 million) during the period of 1 January 2000 to 31 December 2000 by Oracle’s internal random number generator. Likewise, the LHID2005 encompasses 1,000,000 insurance claim data drawn from the year 2005 Registry of Beneficiaries of the NHIRD (25.68 million) during the period of 1 January 2005 to 31 December 2005 following the same method. Through statistic validation, there was no selection bias on age, sex, or insurance premiums between the LHID and the NHIRD. Both the LHID2000 and LHID2005 adopted the 2001 Edition of International Classification of Diseases, Ninth Revision, Clinical Modification (ICD-9-CM). This study was approved by the Institutional Review Board (IRB) of Chang Gung Medical Foundation, and adhered to the tenets of the Declaration of Helsinki. Given the retrospective nature of this study and the use of de-identified patient data in the LHID, the IRB of the Chang Gung Medical foundation approved that the requirements of informed consent were waived.

### 2.2. Study Design

A flowchart of enrollment of obese and non-obese groups is presented in Figure 1. The LHID2000 and LHID2005 were separately reviewed in this matched-cohort study. There were 17,256 and 17,474 individuals aged ≥18 years included from the LHID2000 and LHID2005. The obese individuals were qualified with the coding of obesity or morbid obesity (ICD-9-CM codes 278.0, 278.00, 278.01) between 1 January 1997 and 31 December 2013. We washed out the initial 4-year period to eliminate individuals with incomplete data and those diagnosed with obesity before 2001 (LHID2000/LHID2005: 1889/1880 individuals). Simultaneously, individuals with a new diagnosis of obesity after 2010 were excluded to confirm that individuals can be traced for at least 3 years (LHID2000/LHID2005: 3365/3397 individuals). Further, individuals diagnosed with OAG before a coding of obesity were also excluded (LHID2000/LHID2005: 60/74 individuals). Final numbers of 11,939 and 12,118 individuals with a new diagnosis of obesity between 2001 and 2010 were recruited in obese groups of the LHID2000 and LHID2005. Individuals without a diagnosis of overweight or obesity (ICD-9-CM codes: 278.0, 278.00, 278.01, 278.02, and 278.1) were included in the non-obese group. The obese individuals were matched with the non-obese individuals at a 1:4 ratio by sex, age, urbanized level, and income. The stratification of urbanization level and income followed the criteria reported in a previous LHID study [17]. All individuals were traced from the index date to the diagnosis of OAG, occurrence of death, or 31 December 2013.

To increase the validation of OAG, outcomes were rigorously defined as the coding of OAG (ICD-9-CM codes 365.1, 365.10, 365.11, 365.12), with a treatment involving anti-glaucoma drugs or surgeries ≥2 times, adjudicated by an ophthalmologist(s) over one year. Multiple covariates were collected for adjustment of the risk of OAG: diabetes mellitus (ICD-9-CM codes 250x), hypertension (ICD-9-CM codes 401x–405x), hyperlipidemia (ICD-9-CM codes 272x), ischemic heart disease (IHD) (ICD-9-CM codes 410x–414x), chronic kidney disease (CKD) (ICD-9-CM codes 585x and 586x), myopia (ICD-9-CM codes 367.1), migraine (ICD-9-CM codes 346x), hypothyroidism (ICD-9-CM code 244.9), obstructive sleep apnea (OSA) (ICD-9-CM codes 327.23, 780.51, 780.53, and 780.57), and hypotension (ICD-9-CM codes 458x). The inclusion criteria for covariates were diagnostic coding ≥1 time in admission or ≥3 times in ambulatory visits.

### 2.3. Statistical Analyses

All statistical data in this study were processed and analyzed using SAS version 9.4 (SAS Inc., Cary, NC, USA). For baseline characteristics, a chi-squared test and Student’s *t* test were used to analyze categorical and continuous variables, respectively. The cumulative incidence of OAG was calculated using the Kaplan–Meier method and a log-rank test. A multivariable Cox proportional regression was applied to estimate the adjusted hazard ratio (aHR) and 95% confidence interval (CI) for risk of OAG. The main model for adjustment consisted of demographic variables: sex, age, urbanization level, and income, and five cardinal covariates associated with risk of OAG: diabetes mellitus, hypertension, hyperlipidemia, ischemic heart disease, and chronic kidney disease. Another five potential covariates of OAG—myopia, hypothyroidism, migraine, obstructive sleep apnea, and hypotension—were added one by one into the main model for testing the stability of the sensitivity analysis. Ultimately, sex, age, and covariates were tested for risk of OAG in multivariable stratified analysis. Statistical significance was defined as a two-sided *p* < 0.05.

## 3. Results

In the LHID2000, OAG occurrence was 99 in 11,939 obese individuals, and the incidence rate of OAG was 1.0 and 0.6 per 1000 person-years in the obese and non-obese groups (incidence rate ratio (IRR) 1.79), respectively. In the LHID2005, OAG occurrence was 122 in 12,118 obese individuals, and the incidence rate of OAG was 1.3 and 0.7 per 1000 person-years in the obese and non-obese groups (IRR 1.85), respectively. The average follow-up was 7.91 ± 2.93/7.93 ± 2.93 years in the obese/non-obese groups of the LHID2000, respectively, and 7.88 ± 2.92/7.89±2.92 years in the obese/non-obese groups of LHID2005, respectively. The ratio of females to males was 2:1 in both databases. For age distribution, the ratio of obese individuals aged ≤40 and >40 years was approximately 1:1 in both databases. Obese adults had a higher rate and frequency of ophthalmology visits than non-obese adults. Additionally, obese adults had higher rates of diabetes mellitus, hypertension, hyperlipidemia, ischemic heart disease, chronic kidney disease, myopia, obstructive sleep apnea, migraine, and hypothyroidism than non-obese adults in both databases (Table 1).

Kaplan–Meier curve analysis revealed that the obese group had a significantly higher cumulative incidence of OAG than the non-obese group in both databases (*p* < 0.001) (Figure 2A,B). In the age-stratified analysis, both the young and older obese groups still showed an increased accumulation of OAG compared with the young and older non-obese groups, and the young obese group displayed a remarkable accumulation of OAG in both databases (Figure 2C,D).

The risk of OAG in the multivariable Cox proportional hazard regression are summarized in Table 2. The risk of OAG occurrence was significantly higher in the obese group than in the non-obese group after adjusting for the main model (aHR 1.43 in the LHID2000; aHR 1.54 in the LHID2005). Men appeared to have a higher risk of OAG than women; however, the difference did not reach statistical significance. Older adults (aged >40 years) had an increased risk of OAG compared with young adults (aged ≤40 years). Among covariates, diabetes mellitus was a prominent risk factor for OAG in both databases.

In sensitivity analysis of the risk of OAG (Figure 3), after adjustment for full model or potential covariates, the aHR of OAG occurrence in obesity remained steady, from 1.32 to 1.43 in the LHID2000, and from 1.46 to 1.58 in the LHID2005.

The multivariable stratified analysis is presented in Figure 4. In the age-stratified analysis, there was a demonstrably higher risk in young obese adults than in young non-obese adults, but there was no statistical significance of OAG risk in older adults. In the sex-stratified analysis, obese men had a higher risk of OAG compared with the non-obese men in both databases; however, obese women had an unremarkable (LHID2000) and borderline (LHID2005) risk of OAG. Moreover, adults with at least one ophthalmology visit were analyzed, and the result showed that obese adults still had a higher risk of OAG than non-obese adults in both databases. In the covariate-stratified analysis, the results revealed that in the LHID2005, obese adults without each covariate had a significantly higher risk of OAG than non-obese adults without each covariate, but obese adults with each covariate had a nonsignificant hazard for OAG compared with non-obese adults with each covariate. There was a similar tendency in most analyses of covariates in the LHID2000, although the analysis of hypertension and hyperlipidemia was discordant between the LHID2000 and LHID2005.

## 4. Discussion

In this two-database, matched-cohort study, obese adults had a higher cumulative incidence of OAG at the 13-year follow-up. Overall, obesity was a significant hazard of OAG after adjustment for multivariable covariates. Moreover, young obese adults had a remarkably higher risk of OAG than young non-obese adults.

The cardinal pathogenesis of OAG is the progressive degeneration of RGCs. Several mechanisms could be considered as to how obesity contributes to the damage of RGCs in OAG. Hormonal disequilibrium in obesity plays an integral role in the progressive impairment of RGCs. Decreased adiponectin and increased leptin resistance in obesity result in insulin resistance, dysfunctional lipid metabolism, atherosclerosis, and the activation of proinflammatory cytokines and oxidative stress, which lead to vascular hypo-perfusion and chronic inflammation of RGCs [18,19,20]. OAG is firmly associated with neurodegenerative pathogeneses such as dysregulation of the brain-derived neurotrophic factor (BDNF), phosphorylation of the tau protein, and overexpression of the apolipoprotein E (APOE) gene [21,22,23]. These neurodegenerative pathways have crucial connections with obesity. BDNF is a potential neuroprotective molecule that prevents neuron damage and synaptic disturbances. Recent studies have demonstrated that decreased BDNF has a profound effect on neurodegenerative diseases in the obese population [24]. Furthermore, leptin resistance in obesity plays a critical role in the formation of tau phosphorylation, which may have a detrimental effect on the impairment of RGCs [25]. Meanwhile, the fat-mass and obesity-associated (FTO) gene, strongly related to obesity, was found to be associated with risk of Alzheimer’s disease through interaction with the APOE gene [26], and modulates cholesterol metabolism in the central neural network [23]. It is possible that obesity orchestrates a similar genetic vulnerability in the pathogenesis of OAG.

This was an Asian population-based study used to substantiate the association between obesity and OAG, and the results suggest that obesity is a potential risk factor for OAG, especially in young obese adults. This result, however, contradicts those of some previous population-based studies, in which the increase of BMI was associated with a lower risk of OAG. There are several possible explanations for this divergence. First, we only included truly obese adults, because physicians in Taiwan coded obesity or morbid obesity as patients with a BMI ≥ 30 kg/m^2^ who intended to seek medical assistance (e.g., including dietary consultation, enrollment in a weight-loss program, the use of anti-obesity drugs, or evaluation for bariatric surgery) [27]. Likewise, Newman-Casey’s study, in which the result showed obesity is a risk factor of OAG, also included obese patients as the study population [14]. By contrast, the study by Pasquale and Ramdas included patients with a wide range of BMI values, which were not thoroughly specific for an obese population [15]. In Kim’s study, they adopted the overweight status (BMI > 25 kg/m^2^) to evaluate the risk of OAG instead of the obese status [16]. Body fat is the critical factor that induces a series of physical dysfunctions in obesity. However, increase in BMI cannot truly reflect the increased mass of body fat, especially in adults with who are overweight or have a normal weight [28]. Further, obesity contributes to more severe morbidities and mortality than overweight [29]. Several studies have even reported that overweight is related to lower mortality rates than normal weight [30,31]. This is why our results showed that obesity is a significant hazard of OAG, because this study recruited truly obese adults who were in need of medical assistance.

Second, previous studies have included study populations aged >40 years, whereas our study recruited study populations aged ≥18 years. In Table 2, the risk of OAG was significantly higher in older adults than in young adults (aged ≤40 years), which is compatible with the concept that OAG is a strong age-related neurodegenerative disease. In the age-stratified analysis, obesity was a nonsignificant hazard for OAG in older adults (aged >40 years) although there was an increased risk of OAG in obese adults. This result is similar to most of the previous studies in that obesity was a nonsignificant risk factor for OAG in adults aged >40 years. This outcome could partially determine age as the most conspicuous risk factor for OAG, and show that age-related neurodegenerative effects result in a more profound risk for OAG than obesity-related neurodegenerative effects. Meanwhile, the degeneration of optic neurons is susceptible to long-term chronic diseases such as hypertension and diabetes, and metabolic diseases among older obese adults. These multiple factors would make the impact of obesity inconspicuous when assessing risk of OAG.

On the other hand, our data indicated that obesity is a remarkable contributor to OAG in young adults (aged ≤40 years) regardless of accumulative incidence or age-stratified analysis. This result may have suggested that without the influence of aging and long-term chronic diseases, obesity would be a decisive factor for accelerating the damage of neurons, and that obesity-related neurodegenerative effects have a crucial impact on OAG occurrence among young adults [32]. Moreover, obesity may be correlated with the pathogenesis of juvenile OAG, such as mutation of the CYP1B1 gene or the glaucoma-associated olfactomedin domain of myocilin [33,34]. With obesity trending among younger populations, it is important to raise the awareness that obesity could be a potential risk factor for OAG in young adults.

In this study, obese adults had a higher rate and frequency of ophthalmology visits than non-obese adults, which is reasonable because obese adults with diabetes, hypertension, or metabolic diseases are often asked for the ophthalmic survey of cataract, retinopathy, or macular degeneration. This may have increased the incidental diagnosis of OAG that resulted in a selection bias of higher OAG risk in obese adults. In order to reduce the effect of higher rates of ophthalmology visits among obese adults, we analyzed the risk of OAG in adults with at least one ophthalmology visit, and the data showed that obesity is still a significant risk factor for OAG after multivariable adjustment. In addition, this study showed that the obese populations skewed towards female and young adults. It is assumed that female and young adults may be more concerned with their physical stature and health than male or older adults, so that these two populations are more active in seeking medical assistance to control weight. To reduce sex and age biases, a complete matching strategy was adopted in this study to mitigate the bias of these factors.

The analysis of both databases demonstrated that men exhibit a higher risk for OAG than women, although the HR did not reach statistical significance. This phenomenon could be explained by the fact that estrogen can protect RGCs in women [35]. In the sex-stratified analysis, obese men had a significantly higher HR of OAG than non-obese men. One reason may be that there are decreased adiponectin levels and an increased insulin resistance among obese men compared with non-obese men [36]. On the other hand, the risk of OAG was relatively indistinct between obese and non-obese women, which could imply that elevated circulating estrogen in obese menopausal women may reduce the risk of OAG [37].

In the multivariable analysis, diabetes mellitus showed a strong association with risk of OAG. Probable pathophysiological mechanisms include insulin resistance, vascular dysregulation, and reduction of neurotrophic factors [38]. In the covariate-stratified analysis, obesity without each covariate was shown to be a significant OAG hazard, but obesity with each covariate revealed a nonsignificant risk of OAG in the LHID2005. Most stratified analyses of covariates had a similar tendency in the LHID2000. These results indicated that even in adults without these covariates, obesity would be a prominent risk factor for OAG. Nevertheless, in adults with these covariates, mutual interaction between obesity and covariates may weaken the impact of obesity on OAG occurrence, and obesity could become a relatively inconspicuous risk factor for OAG. Another reason for the nonsignificant hazard of OAG in obese adults with covariates could be due to the small number of cases because the analysis revealed a wide confidence interval. Further, the stratified analysis of hypertension and hyperlipidemia were inconsistent between the LHID2000 and LHID2005, and a larger study population is required to elucidate their role on the association between obesity and OAG in advanced studies.

This study had several limitations. First, the data of BMI were not collected in the LHID, and the dynamic changes of weight could not be traced. Additionally, the obese population may have been underestimated, as most physicians code obesity only if the obese patient wants to seek out medical assistance. On the other hand, the coding of obesity is rather reliable in Taiwan, because physicians are required to code patients with obesity for medical intervention, otherwise they will face the risk of an audit and heavy penalties under the monthly review of the Bureau of National Health Insurance. In addition, the incidence rate of OAG may have been underestimated. In clinical experience, obese adults often experience a long period of obesity before coding; however, we excluded all adults diagnosed with OAG before the coding of obesity could curtail the incidence of OAG in obese adults. Second, IOP is a critical risk factor for OAG, and most studies have already reported that obesity is highly associated with elevated IOP. However, the data of IOP was not collected in the LHID, so the mutual interaction among obesity, IOP, and OAG could not be evaluated without IOP data. Third, the definition of OAG depends on the coding system of the ICD-9. To validate the accuracy of OAG, we strictly defined the outcome as (at least) a coding of OAG with a treatment involving anti-glaucoma drugs or surgeries ≥2 times, adjudicated by an ophthalmologist(s) over one year. Moreover, two independent databases were used to confirm the consistency of the results. Finally, there is no personal history collected in the NHIRD, so the effect of smoking was not taken into account.

To summarize, in a Taiwanese-based population, the results of this two-database matched-cohort study suggested that obese adults have an increased hazard of OAG. In age stratification, obesity could be a potential risk factor of OAG in young adults, but obesity poses a nonsignificant risk for OAG in older adults. In this era of obesity trending among young adults, more attention should be paid to the impact of obesity on OAG occurrence, and a recommendation of ophthalmic survey should be considered in young obese adults, not only for those with metabolic diseases. A comprehensive understanding of the association between obesity and OAG could have a far-reaching influence on the lives of young obese adults. Obesity-related hormonal disequilibrium, neurologic disturbance, and genetic dysregulation are possible pathogenic mechanisms of OAG. Advanced clinical and laboratory research will be essential for elucidating the relationship between obesity and OAG.

## Figures and Tables

**Figure 1 jcm-10-04021-f001:**
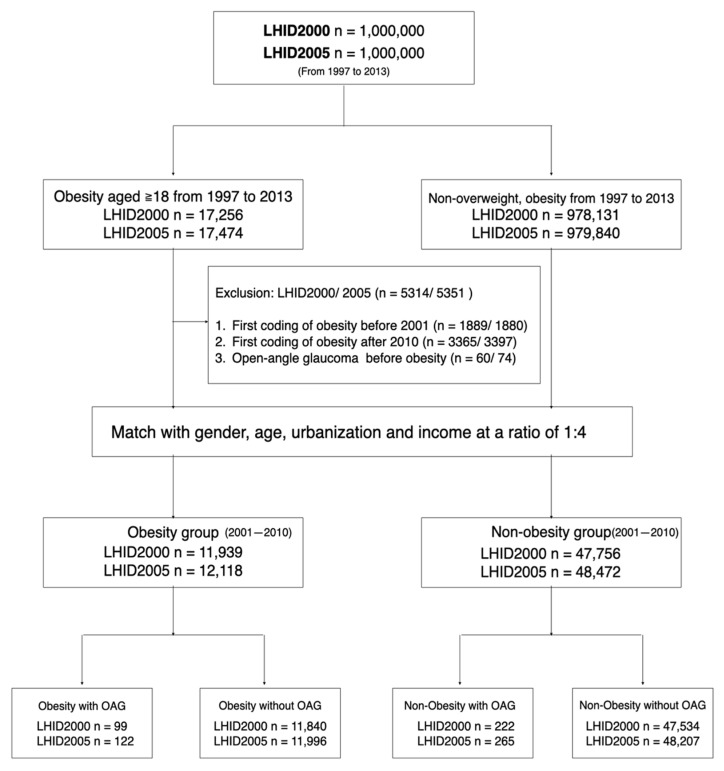
Flowchart of enrollment and allocation of adults with obesity and non-obesity in a two-database matched-cohort study. Obese adults aged ≥18 years in the LHID2000 and LHID2005 between 1 January 1997 and 31 December 2013 were included. Adults first diagnosed with obesity before 2001, adults first diagnosed with obesity after 2010, and adults diagnosed with OAG before obesity were excluded in this study. Obese adults were matched with non-obese adults at a 1:4 ratio by sex, age, urbanization level, and income. All adults were traced until the death, occurrence of OAG, or 31 December 2013. LHID = longitudinal health insurance database; OAG = open-angle glaucoma.

**Figure 2 jcm-10-04021-f002:**
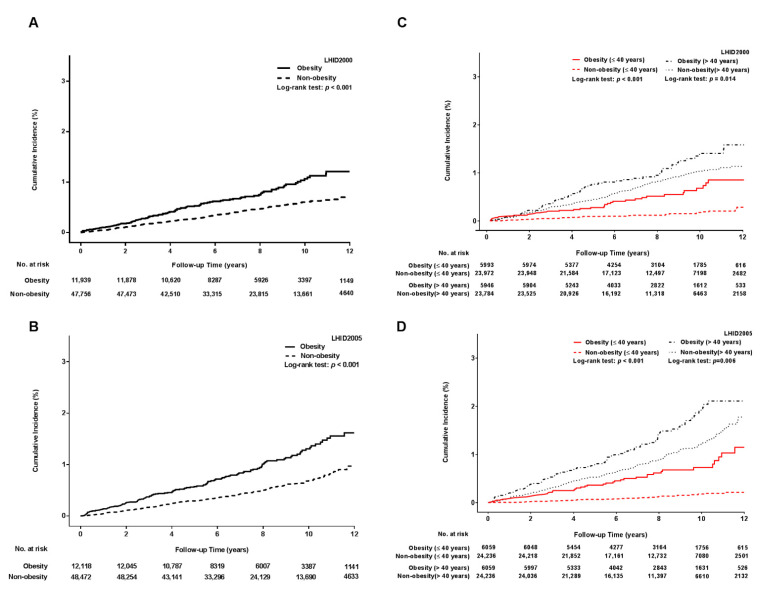
Cumulative incidence of open-angle glaucoma in obese and non-obese adults. (**A**) Cumulative incidence of OAG in the LHID2000; (**B**) cumulative incidence of OAG in the LHID2005; (**C**) age-stratified analysis for the cumulative incidence of OAG in the LHID2000; (**D**) age-stratified analysis for the cumulative incidence of OAG in the LHID2005. Both the LHID2000 and LHID2005 demonstrated that the cumulative incidence of OAG was higher in obese adults than non-obese adults (Log-rank test *p* < 0.001). In the age-stratified analysis, young obese adults (red line) displayed a remarkable cumulative incidence of OAG compared with young non-obese adults (red dotted line) (Log-rank test *p* < 0.001) although both the young and older obese adults had a significantly higher cumulative incidence of OAG. LHID = longitudinal health insurance database; OAG = open-angle glaucoma.

**Figure 3 jcm-10-04021-f003:**
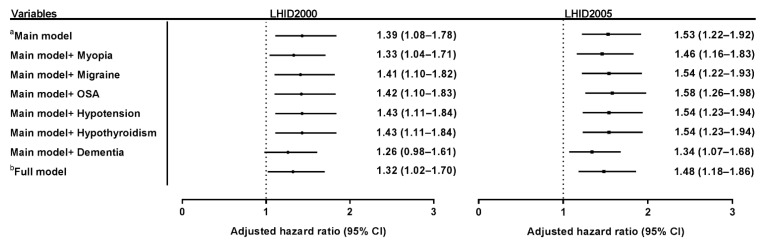
Sensitivity analysis of the risk of open-angle glaucoma in obesity. Sensitivity analysis in both the LHID2000 and LHID2005 showed that the adjusted hazard ratio of OAG remained stable after being adjusted according to the full model and main model with each potential covariate (myopia, migraine, OSA, hypotension, and hypothyroidism). The adjusted hazard ratio of OAG was 1.32 to 1.43 in the LHID2000, and 1.49 to 1.58 in the LHID2005. ^a^ Main model adjusted for age, gender, income, urbanization, diabetes mellitus, hypertension, hyperlipidemia, ischemic heart disease, and chronic kidney disease ^b^ Full model adjusted for the main model, myopia, hypothyroidism, migraine, OSA, and hypotension. OAG = open-angle glaucoma; OSA = obstructive sleep apnea.

**Figure 4 jcm-10-04021-f004:**
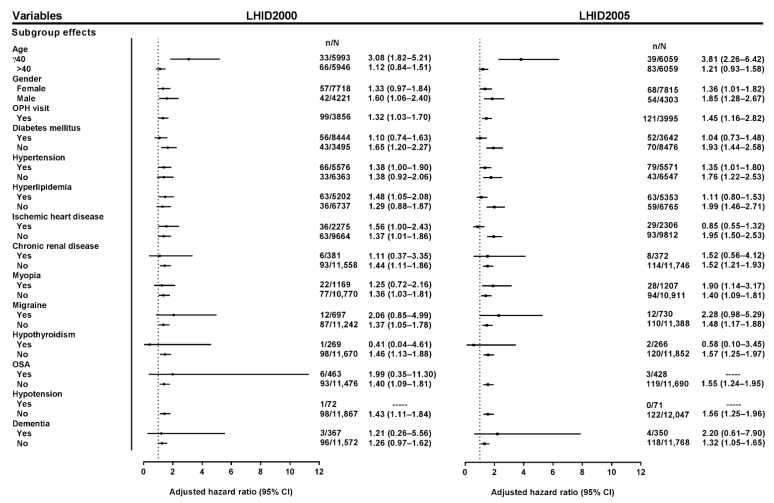
Multivariable stratified hazard analysis of the association between obesity and open-angle glaucoma. All adjusted hazard ratios in the multivariable stratified analysis were adjusted by the main model. The results in both the LHID2000 and LHID2005 displayed that obesity was a remarkable hazard of OAG in young (aged ≤40 years) obese adults compared with young non-obese adults. In gender stratification, the risk of OAG was more significant in obese men than non-obese men. As for the stratified analysis of covariates, in the LHID2005, the obese group without each covariate had an increased risk of OAG compared with the non-obese group without each covariate; however, the obese group with each covariate had a nonsignificant risk of OAG compared with the non-obese group with each covariate. There was a similar tendency in most of the stratified analyses of covariates in the LHID2000, but the analyses of hypertension and hyperlipidemia were inconsistent between the LHID2000 and LHID2005. There was no adjusted hazard ratio in a few subgroups (obesity with OSA in the LHID2005, and obesity with hypotension in both the LHID2000 and LHID2005). LHID = longitudinal health insurance database; OAG = open-angle glaucoma; OSA = obstructive sleep apnea; OPH = ophthalmology.

**Table 1 jcm-10-04021-t001:** Baseline characteristics with incidence of open-angle glaucoma in obese and non-obese adults.

	LHID2000			LHID2005		
	Obesity	Non-Obesity		Obesity	Non-Obesity	
Variables	*n* (%)	*n* (%)	*p*-value	*n* (%)	*n* (%)	*p*-value
Total	11,939	47,756		12,118	48,472	
Gender			1.00			1.00
Female	7718 (64.6)	30,872 (64.6)		7815 (64.5)	31,260 (64.5)	
Male	4221 (35.4)	16,884 (35.4)		4303 (35.5)	17,212 (35.5)	
Age			1.00			1.00
18–30	3135 (26.3)	12,540 (26.3)		3098 (25.6)	12,392 (25.6)	
31–40	2858 (23.9)	11,432 (23.9)		2961 (24.4)	11,844 (24.4)	
41–50	2660 (22.3)	10,640 (22.3)		2645 (21.8)	10,580 (21.8)	
51–60	1992 (16.7)	7968 (16.7)		2054 (17.0)	8216 (17.0)	
61–70	886 (7.4)	3544 (7.4)		944 (7.8)	3776 (7.8)	
>70	408 (3.4)	1632 (3.4)		416 (3.4)	1664 (3.4)	
Urbanization level			1.00			1.00
1(City)	3965 (33.2)	15,860 (33.2)		3765 (31.1)	15,060 (31.1)	
2	5477 (45.9)	21,908 (45.9)		5837 (48.2)	23,348 (48.2)	
3	1673 (14.0)	6692 (14.0)		1746 (14.4)	6984 (14.4)	
4(Villages)	824 (6.9)	3296 (6.9)		770 (6.4)	3080 (6.4)	
Income			1.00			1.00
0	3952 (33.1)	15,808 (33.1)		2447 (20.2)	9788 (20.2)	
1–15,840	1883 (15.8)	7532 (15.8)		1811 (14.9)	7244 (14.9)	
15,841–25,000	3990 (33.4)	15,960 (33.4)		3742 (30.9)	14,968 (30.9)	
≥25,001	2114 (17.7)	8456 (17.7)		4118 (34.0)	16,472 (34.0)	
OPH visit	3856 (32.3)	12,082 (25.3)	<0.001	3995 (33.0)	12,731 (26.3)	<0.001
Frequency of OPH visit			<0.001			<0.001
0	8083 (67.7)	35,674 (74.7)		8123 (67.0)	35,741 (73.7)	
1–2	2458 (20.6)	8061 (16.9)		2564 (21.2)	8470 (17.5)	
>2	1398 (11.7)	4021 (8.4)		1431 (11.8)	4261 (8.8)	
Comorbidity						
Diabetes mellitus	3494 (29.3)	5419 (11.4)	<0.001	3642 (30.1)	5810 (12.0)	<0.001
Hypertension	5576 (46.7)	10,818 (22.7)	<0.001	5571 (46.0)	11,434 (23.6)	<0.001
Hyperlipidemia	5202 (43.6)	8326 (17.4)	<0.001	5353 (44.2)	8939 (18.4)	<0.001
IHD	2275 (19.1)	4510 (9.4)	<0.001	2306 (19.0)	4867 (10.0)	<0.001
CKD	381 (3.2)	969 (2.0)	<0.001	372 (3.1)	1027 (2.1)	<0.001
Myopia	1169 (9.8)	3422 (7.2)	<0.001	1207 (10.0)	3562 (7.4)	<0.001
Migraine	697 (5.8)	1618 (3.4)	<0.001	730 (6.0)	1815 (3.7)	<0.001
Hypothyroidism	269 (2.3)	354 (0.7)	<0.001	266 (2.2)	404 (0.8)	<0.001
OSA	463 (3.9)	319 (0.7)	<0.001	428 (3.5)	313 (0.7)	<0.001
Hypotension	72 (0.6)	225 (0.5)	0.067	71 (0.6)	234 (0.5)	0.151
No. of OAG	99	222		122	265	
Incidence %	1.0	0.6	<0.001	1.3	0.7	<0.001
IRR (95% CI)	1.79 (1.41–2.27)			1.85 (1.49–2.29)		

CI = confidence interval; CKD = chronic kidney disease; IHD = ischemic heart disease; LHID = Longitudinal Health Insurance Database OAG = open-angle glaucoma; OSA = obstructive sleep apnea; OPH = ophthalmology IRR = incidence rate ratio.

**Table 2 jcm-10-04021-t002:** Multivariable cox proportional hazard regression of the association between open-angle glaucoma and potential risk factors.

	LHID2000				LHID2005			
Variables	Crude HR (95% CI)	*p*-Value	aHR (95% CI)	*p*-Value	Crude HR (95% CI)	*p*-Value	aHR (95% CI)	*p*-Value
Exposure								
Non-obesity	1 (Reference)		1 (Reference)		1 (Reference)		1 (Reference)	
Obesity	1.79 (1.41–2.27)	<0.001	1.43 (1.11–1.84)	0.006	1.85 (1.49–2.29)	122	1.54 (1.23–1.94)	<0.001
Gender								
Female	1 (Reference)		1 (Reference)		1 (Reference)		1 (Reference)	
Male	1.23 (0.98–1.54)	0.076	1.21 (0.95–1.52)	0.120	1.16 (0.95–1.43)	0.153	1.19 (0.96–1.47)	0.109
Age								
Age ≤ 40	1 (Reference)		1 (Reference)		1 (Reference)		1 (Reference)	
Age > 40	3.83 (2.94–5.00)	<0.001	3.09 (2.26–4.25)	<0.001	4.75 (3.66–6.15)	<0.001	3.52 (2.64–4.70)	<0.001
Comorbidity								
Diabetes								
No	1 (Reference)		1 (Reference)		1 (Reference)		1 (Reference)	
Yes	2.91 (2.31–3.66)	<0.001	1.55 (1.18–2.04)	0.002	3.04 (2.47–3.73)	<0.001	1.61 (1.26–2.06)	<0.001
Hypertension								
No	1 (Reference)		1 (Reference)		1 (Reference)		1 (Reference)	
Yes	2.68 (2.16–3.34)	<0.001	1.22 (0.92–1.61)	0.170	3.00 (2.46–3.67)	<0.001	1.38 (1.07–1.77)	0.012
Hyperlipidemia								
No	1 (Reference)		1 (Reference)		1 (Reference)		1 (Reference)	
Yes	2.51 (2.01–3.12)	<0.001	1.14 (0.87–1.50)	0.347	2.27 (1.86–2.78)	<0.001	0.90 (0.71–1.15)	0.412
IHD								
No	1 (Reference)		1 (Reference)		1 (Reference)		1 (Reference)	
Yes	2.66 (2.08–3.42)	<0.001	1.27 (0.95–1.68)	0.106	2.60 (2.07–3.25)	<0.001	(1.170.91–1.50)	0.231
CKD								
No	1 (Reference)		1 (Reference)		1 (Reference)		1 (Reference)	
Yes	2.17 (1.29–3.64)	0.003	1.02 (0.60–1.74)	0.934	2.05 (1.28–3.30)	0.003	0.93 (0.57–1.51)	0.764

CKD = chronic kidney disease; IHD = ischemic heart disease; IRR = incidence rate ratio; LHID = Longitudinal Health Insurance Database OAG = open-angle glaucoma.

## Data Availability

The data presented in this study are available on request from the corresponding author. The data are not publicly available due to privacy and ethical reasons.

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
