# Peer review of "Is Obesity a Risk or Protective Factor for Open-Angle Glaucoma in Adults? A Two-Database, Asian, Matched-Cohort Study"

_jcm, 2021, doi:10.3390/jcm10174021_

Round 1

Reviewer 1 Report

It seems to me to be an interesting and statistically well-conducted study. However, I believe that the authors do not fully appreciate the limitations of a retrospective study, in which the patients come from a database far from a close and personal clinical work. For this reason, I also believe that conclusions and hypotheses are drawn that could be far from reality. Therefore, I consider that the paper would deserve to be published if the results were discussed from a different perspective, based on the paradigm that "coincidence does not necessarily mean causality".

Let me explain: It is well known that, even in the most developed countries, most glaucoma goes undiagnosed. Often throughout the patient's life. Therefore, observing a greater number of glaucomatous disease among patients who come for health check-ups for another type of pathology does not necessarily mean that the incidence is higher in that population, much less that there is a cause-effect relationship between the two. This has been much discussed, for example, in the case of myopes or diabetics, whose ophthalmologic care is much more frequent than that of the general population.

The same is true for obesity. Maybe not so much directly but because of its association with diabetes or arterial hypertension. These types of patients may have been diagnosed with glaucoma more than those who are not overweight, simply because they had more intense medical care. Thus, in 2007 Cheung et al (Surv Ophthalmol. 2007 ; 52(2): 180-195), who are not cited by the authors of the present paper, conducted an extensive review on this topic and indicated that there were no solid arguments to defend a causal association between obesity and glaucoma. Another meta-analysis, also not cited by the authors, would support their hypothesis (Liu et al Journal of Ophthalmology. Hindawi. 2017, 9787450. https://doi.org/10.1155/2017/9787450).

On the other hand, it is understandable to establish pathogenic hypotheses on a racial basis when no other more obvious explanation can be found. However, in this case it does not seem justified. The authors assume that the differences with respect to what has been described in other studies in Western populations may be due to specific metabolic characteristics of the Asian population. However, they ignore the paper of Kim et al (Ophthalmology, 2016:1-10), who found in Korean population just the opposite of what they refer in their paper. Specifically they report that "overweight status reduced the risk of POAG". It is quite possible that this relationship may have also been coincidental or the product of some methodological bias, but it is clearly contradictory to the result presented in the current paper.

In the last paper I have pointed out, the authors indicate, as in many other studies, that "only 8.0% of the POAG patients previously had been reported and were aware of the disease". Indeed, a large proportion of glaucomatous patients are unaware of the disease. Similar explanations may justify the differences observed in glaucoma with respect to sex and other variables. For example, the greater tendency of women in some cultures to be more attentive to their health than men. It would not be surprising if the opposite were true in other cultures.

The study would only be valid if all subjects were examined by ophthalmologists in order to identify or exclude glaucoma, but not because this diagnosis appears or does not appear in their clinical history.

For all these reasons, I believe that the authors should reduce their expectations of explaining the results with medical, metabolic or racial arguments and make an effort to introduce other possible sociological reasons that may have caused them. From a health point of view, these could be as important, or even more important, than the ones they suggest.

Author Response

It seems to me to be an interesting and statistically well-conducted study. However, I believe that the authors do not fully appreciate the limitations of a retrospective study, in which the patients come from a database far from close and personal clinical work. For this reason, I also believe that conclusions and hypotheses are drawn that could be far from reality. Therefore, I consider that the paper would deserve to be published if the results were discussed from a different perspective, based on the paradigm that "coincidence does not necessarily mean causality".

Let me explain: It is well known that, even in the most developed countries, most glaucoma goes undiagnosed. Often throughout the patient's life. Therefore, observing a greater number of glaucomatous diseases among patients who come for health check-ups for another type of pathology does not necessarily mean that the incidence is higher in that population, much less that there is a cause-effect relationship between the two. This has been much discussed, for example, in the case of myopes or diabetics, whose ophthalmologic care is much more frequent than that of the general population.

Reply: Thanks for your valuable comment. We agree that obese adults with diabetes, hypertension, or metabolic diseases are often asked for the ophthalmic survey of cataract, retinopathy, or macular degeneration, and it is possible to increase the accidental diagnosis of OAG while ophthalmic survey and make a bias of high OAG risk in obese adults. Indeed, we found that obese adults in this study had a higher rate and frequency of ophthalmology visits than non-obese adults, and we already presented the data in Table 1. In order to reduce this bias of higher rate of ophthalmology visits in obese adults, we analyzed the risk of OAG in adults with at least one ophthalmology visit, and the data showed that obesity is still a significant hazard of OAG after multivariable adjustment. We also presented this data in Fig 4. In addition, we already delineated the concern in the discussion section (from Line 320 to 328).  

The same is true for obesity. Maybe not so much directly but because of its association with diabetes or arterial hypertension. These types of patients may have been diagnosed with glaucoma more than those who are not overweight, simply because they had more intense medical care. Thus, in 2007 Cheung et al (Surv Ophthalmol. 2007; 52(2): 180-195), who are not cited by the authors of the present paper, conducted an extensive review on this topic and indicated that there were no solid arguments to defend a causal association between obesity and glaucoma. Another meta-analysis, also not cited by the authors, would support their hypothesis (Liu et al Journal of Ophthalmology. Hindawi. 2017, 9787450. https://doi.org/10.1155/2017/9787450).

Reply: Thanks for your kind suggestion. We understand that obesity is associated with multiple systemic diseases such as diabetes, hypertension, hyperlipidemia, ischemic heart disease, and chronic renal disease that would increase the risk of OAG. Hence, in this study, we already collected associated covariates to adjust the effect of obesity on the risk of OAG. The review study of Cheung et al (Surv Ophthalmol. 2007; 52(2): 180-195) primarily suggested Obesity is highly associated with increased IOP, but the association between obesity and glaucoma was inconclusive. This point was already delineated in our introduction and we already cited the paper. On the other hand, the meta-analysis of study (Liu et al Journal of Ophthalmology. Hindawi. 2017, 9787450) primarily discussed the association between adiposity and the risk of glaucoma, and concluded that adiposity has a higher risk of elevated IOP, and abdominal adiposity has a positive association with glaucoma. However, there is no statistical significance between adiposity and the risk of open-angle glaucoma. We also cited this study in the introduction section. (At Line 54)

On the other hand, it is understandable to establish pathogenic hypotheses on a racial basis when no other more obvious explanation can be found. However, in this case, it does not seem justified. The authors assume that the differences with respect to what has been described in other studies in Western populations may be due to specific metabolic characteristics of the Asian population. However, they ignore the paper of Kim et al (Ophthalmology, 2016:1-10), who found in Korean population just the opposite of what they refer in their paper. Specifically they report that "overweight status reduced the risk of POAG". It is quite possible that this relationship may have also been coincidental or the product of some methodological bias, but it is clearly contradictory to the result presented in the current paper.

Reply: Thanks for your useful comment. We deleted the discussion of racial bias following your suggestion and focused on the discussion of the contradictory results between our study and previous studies. We cited the study of Kim et al (Ophthalmology, 2016:1-10) and discussed the different results between previous studies and our study. Primarily, we made two points: study population and age issue to explain the different outcomes. In the study population, Studies by Pasquale and Ramdas included patients with a wide range of BMI values, which were not thoroughly specific for an obese population. In Kim’s study, they adopted the overweight status (BMI of > 25) to evaluate the risk of OAG instead of the obese status. However, we included truly obese adults because physicians in Taiwan code the obesity or morbid obesity for patients with BMI ≥ 30 kg/m2 who intended to seek medical assistance. Likewise, Newman-Casey’ study, in which the result showed obesity is a risk factor of OAG, also included obese patients as the study population. Obesity contributes to more severe morbidities and mortality than overweight. Several studies even reported that overweight (BMI of 25-<30) was related to lower mortality rates than normal weight. This is why our result showed that obesity is a significant hazard of OAG because this study recruited truly obese adults with the necessity of medical assistance. In the age argument, obesity was a nonsignificant hazard for OAG in older adults (aged > 40 years) although there was an increased risk of OAG in obese adults. This result is similar to most of the previous studies that obesity is a nonsignificant risk factor of OAG in adults aged > 40 years. Instead, our data indicated that obesity was a remarkable hazard of OAG in young adults (aged ≤ 40 years). This outcome could be explained that age is the most conspicuous risk factor of OAG and age-related neurodegenerative effect is more profound on the risk of OAG than obesity-related neurodegenerative effect. Meanwhile, the degeneration of optic neurons is susceptible to long-term chronic diseases such as hypertension, diabetes, and metabolic diseases among obese older adults. These multiple factors would make the impact of obesity become inconspicuous on the risk of OAG. In young adults, without the influence of aging and long-term chronic diseases, obesity would be a decisive factor of accelerating the damage of neurons, and the obesity-related neurodegenerative effect has a crucial impact on OAG occurrence. We already delineated these arguments in the discussion section (from Line 274 to 319)

In the last paper I have pointed out, the authors indicate, as in many other studies, that "only 8.0% of the POAG patients previously had been reported and were aware of the disease". Indeed, a large proportion of glaucomatous patients are unaware of the disease. Similar explanations may justify the differences observed in glaucoma with respect to sex and other variables. For example, the greater tendency of women in some cultures to be more attentive to their health than men. It would not be surprising if the opposite were true in other cultures.

Reply: Thanks for your kind comment. We actually noticed that a high ratio of female and young obese adults was observed in this study. It is assumed that female and young adults may be more concerned with their physical stature and health than male and older adults, so that they are more active in seeking medical assistance for weight loss. Hence, in order to reduce sex and age bias, we adopted a complete matching strategy of age and gender in this study to mitigate the bias on these factors. We already delineated this issue in the discussion section. (From Line 328 to 333)

The study would only be valid if all subjects were examined by ophthalmologists in order to identify or exclude glaucoma, but not because this diagnosis appears or does not appear in their clinical history.

Reply: Thanks for your comment. We tried to analyze the risk of OAG in adults with at least one ophthalmology visit, and the data showed that obesity is still a significant hazard of OAG after multivariable adjustment. We also presented this data in Fig 4. In addition, we already delineated the concern in the discussion section. (From Line 320 to 328)

For all these reasons, I believe that the authors should reduce their expectations of explaining the results with medical, metabolic, or racial arguments and make an effort to introduce other possible sociological reasons that may have caused them. From a health point of view, these could be as important, or even more important, than the ones they suggest.

Reply: Thanks for your useful suggestion. We already reduced a large part of metabolic, racial, and medical explanations for the result. We just kept more solid evidence including hormonal disequilibrium, neurodegeneration, and genetics related to obesity and OAG in one paragraph. Following your suggestion, we focused on discussing our results on the perspectives of age, gender, study population, ophthalmology visit, and multiple covariates.  

Reviewer 2 Report

The authors posed an interesting question of whether obesity is an independent risk factor for glaucoma and to answer the question they have completed a two-database-matched-cohort study including subjects over 18 years of age and a follow-up period of 13 years. According to their results, subjects with a code for obesity are at increased risk for OAG. After multiple adjustments for possible covariates, this association has remained only in young subjects less than 40 years of age. 

The study design and the presentation of the results are solid. Minor considerations before acceptance of the study are the following:

  1. The authors were unable to adjust their results for IOP. The association of obesity with OAG could be mediated by the possible association of obesity with the IOP. However, data on IOP was not available in the databases.  data on IOP is not useful for the OAG diagnosis as stated by the authors, but as the most important risk factor for OAG. The authors should clearly comment on this in the limitation section of the discussion.
  2. Since the association of obesity with OAG was statistically significant only in the younger group, it will be interesting to report, if possible, the mean time interval between the first coding for obesity and the first coding for OAG in the two age groups. Differences in this time could possibly explain the different associations of obesity with OAG. 

Author Response

The authors posed an interesting question of whether obesity is an independent risk factor for glaucoma and to answer the question they have completed a two-database-matched-cohort study including subjects over 18 years of age and a follow-up period of 13 years. According to their results, subjects with a code for obesity are at increased risk for OAG. After multiple adjustments for possible covariates, this association has remained only in young subjects less than 40 years of age. 

The study design and the presentation of the results are solid. Minor considerations before acceptance of the study are the following:

  1. The authors were unable to adjust their results for IOP. The association of obesity with OAG could be mediated by the possible association of obesity with the IOP. However, data on IOP was not available in the databases.  data on IOP is not useful for the OAG diagnosis as stated by the authors, but as the most important risk factor for OAG. The authors should clearly comment on this in the limitation section of the discussion.

   Reply: Thanks for your valuable suggestion. We agree that IOP is the most important risk factor for OAG. In addition, most studies already reported obesity is highly associated with increased IOP. However, the data of IOP was not collected in the LHID. Hence, we already delineated this limitation in the discussion, and the sentences are that” IOP is a critical risk factor for OAG, and most studies already reported obesity is highly associated with elevated IOP. However, the data of IOP was not collected in the LHID, so that the mutual interaction between obesity, IOP and OAG could not be evaluated without IOP data.” (From Line 368 to 372)

  1. Since the association of obesity with OAG was statistically significant only in the younger group, it will be interesting to report, if possible, the mean time interval between the first coding for obesity and the first coding for OAG in the two age groups. Differences in this time could possibly explain the different associations of obesity with OAG. 

Reply: Thanks for your kind suggestion. We analyzed the mean time interval between the first coding for obesity and the first coding for OAG in the two age groups. The data was shown below

The time interval between the first coding of obesity and the first coding of OAG

Variables

Age≦40

Age>40

P-value*

Fw-time(M±SD)-LHID2000

4.76±3.31 (N=33)

4.77±3.25 (N=66)

0.983

Fw-time(M±SD)-LHID2005

5.06±3.32 (N=39)

4.58±3.09 (N=83)

0.438

*Student’s t tests

The result showed that there was no significant difference in the time interval from the diagnosis of obesity to the diagnosis of OAG between the young obese group and the older obese group. In this cohort study, we can explicitly confirm the sequential order between the occurrence of obesity and OAG. However, it is hard to truly reflect the real-time interval between the obesity and the OAG occurrence because we cannot confirm how long the adults are obese before the first coding for obesity.

Round 2

Reviewer 1 Report

----